# Secondary Rhinaria Contribute to Major Sexual Dimorphism of Antennae in the Aphid *Semiaphis heraclei* (Takahashi)

**DOI:** 10.3390/insects14050468

**Published:** 2023-05-16

**Authors:** Ke-Xin Song, Jiang-Yue Wang, Hai-Li Qiao, Hong-Shuang Wei, Kun Guo, Chang-Qing Xu

**Affiliations:** 1Institute of Medicinal Plant Development, Chinese Academy of Medical Sciences/Peking Union Medical College, Beijing 100193, China; 2College of Life Sciences, Shaanxi Normal University, Xi’an 710119, China

**Keywords:** antennal sensilla, scanning electron microscopy, male, sexual female, ultramorphology

## Abstract

**Simple Summary:**

The host-alternating aphid *Semiaphis heraclei* (Hemiptera: Aphididae) is the dominant pest of its primary host plant—honeysuckle (*Lonicera japonica*). As the initial generation on honeysuckles each year, the sexual generation is a key target in aphid control. Understanding the mechanism underlying chemical communication between sexual aphids could be helpful for exploring olfactory stimulus-based male trapping techniques. In this study, the morphology of antennae and the types, sizes, numbers, and distribution of sensilla on the male and sexual female antennae of *S. heraclei* were investigated by means of scanning electron microscopy. The male antennae were dramatically longer than those of sexual females and the flagellum length contributed to the different antennal length between the two sexes. Four of five sensillum types or subtypes present in both sexes were enlarged in males. In addition, trichoid sensilla subtype I were more abundant on male antennae than on sexual female antennae. In particular, secondary rhinaria were found to be male specific. These sex-biased traits could be of great importance in the perception of sex pheromones. Our findings provide insight into the olfactory sensing mechanism of sexual individuals to assist in pest control.

**Abstract:**

Sexual generation is an important generation in the life cycle of host-alternating aphids, and its population size determines the intensity of the peak in the next spring. Although male trapping techniques based on olfactory stimuli have been successfully established in the field, the biological basis of olfactory perception in males is unclear. In this study, we compared the morphology of antennae and the types, sizes, numbers, and distribution of sensilla between males and sexual females in the host-alternating aphid *Semiaphis heraclei* (Hemiptera: Aphididae). We found that flagellum length differentiation contributed to the majority of the sexual dimorphism of antennae. Most sensillum types or subtypes, including trichoid sensilla subtype I, campaniform sensilla, and primary rhinaria subtypes I and II, were enlarged in males. In addition, males bore more trichoid sensilla subtype I than sexual females. In particular, secondary rhinaria were present in males only and could not be detected in sexual females. These results revealed the structural basis of male olfactory perception. Our findings provide insight into the mechanism underlying chemical communication between sexual aphids and could thus be useful for pest control.

## 1. Introduction

Aphids are devastating pests of agricultural crops. They not only cause direct damage by feeding on leaves and shoots, but also transmit plant viruses that cause destructive damage. Most agriculturally important aphids are host-alternating species, such as *Aphis gossypii* and *Myzus persicae* because their secondary hosts are usually herbaceous species and crop plants [1]. Aphids typically reproduce by cyclical parthenogenesis, alternating one bisexual generation with a succession of parthenogenetic, all-female generations [1]. Most attention has been given to the parthenogenetic females because they are the most common morphs on secondary host plants during the spring/summer. In contrast, sexual individuals usually appear in autumn when environmental conditions start to deteriorate and have not been studied much. Considering their occurrence at specific times and small population size, the sexual generation, as a driver of overwintering egg number and spring population size in the coming year, is an alternative target in pest control [2].

The limited research on sexual individuals has focused on morphology [3,4,5], ecological conditions for production [6,7], and reproductive biology [8,9]. The most studied subjects are the chemical isolation, identification, synthesis, and application of sex pheromones of aphids [10,11,12]. Sex pheromone-based techniques have been used to successfully trap males [13,14], which suggests that males are sensitive to and selective toward species-specific sex pheromones. However, the biological basis of olfactory perception in males is still unclear. In insects, the antennae are the main olfactory organs for sensing chemical information. Antennal sensilla are specialized components of the antennal cuticles and function as organs detecting both external and internal stimuli to regulate behavior via the nervous system [15]. Although the morphology of antennae and sensilla has been described in a few species, such as *M. persicae*, *Acyrthosiphon pisum*, *Sitobion avenae* with both viviparous and bisexual morphs [16,17,18], *A. gossypii* and *A. glycine* with viviparous morphs and the male only [19,20], studies focusing on the sexual differentiation of antennae in males compared with females are lacking.

The host-alternating aphid *Semiaphis heraclei* (Takahashi) (Hemiptera: Aphididae) is an economically important pest that uses *Lonicera* spp. as its primary host plant and Apiaceae plants as secondary host plant [21]. Both primary and secondary host plants are of great importance, especially the flower of honeysuckle (*Lonicera japonica*, the primary host), which is a Chinese traditional medicine that plays an important role in the treatment of SARS-CoV-2 and COVID-19 [22,23]. *S. heraclei* is the dominant pest of honeysuckle [24,25]. In honeysuckle-producing areas, the aphids overwinter on honeysuckle as diapausing eggs. Fundatrices hatch in early spring, and then reproduce parthenogenetically on honeysuckle. In early summer, alate virginoparae migrate to secondary plants and in late autumn, gynoparae and males fly back to honeysuckle, with the former producing sexual females. Males and sexual females mate and lay eggs [21,26]. During April–May, the occurrence peak of this aphid coincides with the flowering period of honeysuckle, affecting flower yields [26,27]. There is a close relationship between sexual population size and the parthenogenetic counterpart on honeysuckle in the spring and investigating the olfactory mechanism used by males to locate sexual females on honeysuckle is critical for aphid control. However, there have been a few studies on the olfactory perception of *S. heraclei*, and information on its antennae and antennal sensilla remains limited.

In the present study, we determined the morphology of antennae and the sensillum types of males and sexual females in *S. heraclei*. We revealed that antennal length, and sensillum size and number were increased in males. Secondary rhinaria were present in males only but could not be detected in sexual females. Our data provide insight for understanding the olfactory perception mechanism of the sexual generation for future pest control.

## 2. Materials and Methods

### 2.1. Experimental Insects

In November 2021, 3rd- or 4th- instar male and sexual female nymphs of *S. heraclei* were collected from a honeysuckle field (116.40° E, 39.93° N) in Beijing, China. The nymphs were then transferred to the laboratory and reared on the honeysuckle cutting seedlings (*L. japonica*) until eclosion. The newly emerged adults were transferred onto new honeysuckle seedlings and reared for 2–5 days before examination. Both nymphs and adults were kept in climatic chambers (PRX-450, Saifu, Ningbo, China) at 12 ± 1 °C and 70 ± 5% relative humidity (RH) with a photoperiod of 8 h:16 h (L:D).

### 2.2. Sample Observation, Photography and Morphometric Measurement

#### 2.2.1. Light Microscopy

Adult males and sexual females were anesthetized with ethyl acetate separately and transferred to honeysuckle leaves. The whole body and the antennae were observed and photographed using a stereoscopic microscope (M205 C, Leica, Wetzlar, Hessian, Germany). All photos were processed with Photoshop software (2021, Adobe, San Jose, CA, USA).

#### 2.2.2. Scanning Electron Microscopy

A total of 15 male and 15 sexual female individuals of *S. heraclei* were subjected to scanning electron microscopy observation. The antennae with the whole body were cleaned three times with 75% ethanol, each time for 5 s in an ultrasonic bath (BL6-180C, Peilai Beijing, Beijing, China), and then fixed in 2.5% glutaraldehyde (EM Grade, Solarbio, Beijing, China) for 12 h at 4 °C. Then, the samples were dehydrated in an incremental ethanol series (30%, 50%, 70%, 90%, and absolute ethanol) for 20 min each. After dehydration, the samples were dried in a critical point dryer (Autosamdri-815, Series A, Tousimis, Rockville, MD, USA) for 3 h and mounted on stubs with double-sided conductive glue to allow the imaging of dorsal, ventral, and lateral views of the antennae. Finally, after being coated with gold in a sputter-coater (MC1000, Hitachi, Tokyo, Japan), the samples were observed and photographed using a scanning electron microscope (SU8020, Hitachi, Tokyo, Japan).

### 2.3. Data Collection and Statistics

The morphometrics were measured with Digimizer software (MedCalc Software, Ostend, Belgium). The measurements were expressed as mean ± standard error. Differences in antenna length and sensillum size and number between sexes were analyzed using *t* test with SPSS statistics (20.0, IBM, Chicago, IL, USA). Bar graphs were drawn with GraphPad Prism 9 software (GraphPad Software Inc. San Diego, CA, USA).

## 3. Results

### 3.1. Flagellum Length Differentiation Contributed the Majority of Sexual Dimorphism of Antennae

We first determined the gross morphology of the antennae of male and sexual female adults (Figure 1A–D) and found that each antenna of both sexes consisted of a cylindrical scape (SC), a cylindrical pedicel (PE), and a slender flagellum including four flagellomeres (FL1–FL4). The last flagellomere (FL4) could be divided into a base part (BA) and a processus terminalis (PT). The surface of the antenna was scaly with various sensilla (Figure 1E–H).

There was a significant difference in antennal length between sexes (T = 24.92, *p* < 0.01). The length of antennae in males (1288.33 µm ± 24.28 µm) was 1.90 times that in females (677.22 µm ± 3.44 µm) (Figure 1I). The flagellum was the largest part of the antennae and accounted for 91.44% and 84.62% of the total antennal length of males and sexual females, respectively. The flagellum length of males was 2.06 times that of sexual females (Figure 1J). Moreover, each flagellomere’s length was significantly longer in males than sexual females (T = 42.93, *p* < 0.01 for FL1; T = 20.94, *p* < 0.01 for FL2; T = 11.71, *p* < 0.01 for FL3; T = 12.01, *p* < 0.01 for FL4) (Figure 1K). Therefore, the difference in flagellum length contributed to the majority of the sexual dimorphism of antennae.

### 3.2. Types and Morphology of Antennal Sensilla

After observing the antennae, we defined the type and morphology of antennal sensilla. According to the morphological characters, there were two single sensillum types, trichoid sensillum and campaniform sensillum, and two types of functional rhinaria, primary rhinarium and secondary rhinarium on the antennae of *S. heraclei* (Figure 2). Trochoid sensilla and primary rhinaria were further divided into two subtypes (Figure 2).


*Trichoid sensilla*


Trichoid sensilla, characterized as hair-like, were located inside the socket on the antennal surface. According to the shape, trichoid sensilla were divided into two subtypes, trichoid sensillum subtype I and trichoid sensillum subtype II. Trichoid sensilla subtype I were slender and lightly curved, with a narrowed tip. Their surfaces were smooth without vertical stripes (Figure 3A,B; Table 1).

Trichoid sensilla subtype II were short and thick, with blunt round ends showing fissure-like structures and grooves (Figure 3C,D; Table 1).


*Campaniform sensilla*


The epidermis of each campaniform sensillum was a hemispherical protrusion with a button-shaped convex formation in the center, surrounded by a smooth and thickened edge (Figure 3E,F; Table 1).


*Primary rhinaria*


According to the constitution, primary rhinaria were divided into two subtypes, primary rhinarium subtype I and primary rhinarium subtype II. Each primary rhinarium subtype I consisted of a single multiparous placoid sensillum surrounded by cuticular projections (Figure 3G; Table 1). The placoid sensillum was an oval plate-like structure located in the lumen with two protrusions on the multiparous surface (Figure 3H; Table 1).

Each primary rhinarium subtype II was composed of one large placoid sensillum, two small placoid sensilla, and four coeloconic sensilla (Figure 3I; Table 1). The placoid sensillum morphology of primary rhinarium subtype II was similar to that of primary subtype I, while there were three protrusions on the surface of the large placoid sensillum (Figure 3J) and one on the small placoid sensillum (Figure 3K,L; Table 1). The epidermis of the coeloconic sensillum was sunken into a circular cavity, the center of which had a sensory cone (Figure 3M,N; Table 1). The top of the cone was a finger-like process formed by cuticular projections, and the edge of the circular cavity had petal-like cuticular projections curved toward the center. Of the four coeloconic sensilla, the apical finger-like process of two sensory cones radiated outward, and the other two were closed (Figure 3M,N).


*Secondary rhinaria*


Each secondary rhinarium was plate-like, with a multiparous placoid sensillum and pronounced on the epidermis of the antenna. The width from the base to the end was basically the same, the end was bluntly rounded, and the surface was covered with pores (Figure 3O,P; Table 1).

### 3.3. Sensilla Sizes Were Universally Enlarged in Males

After defining the antennal sensillum types, we measured the sensillum length or diameter of each type. Among the five sensillum types or subtypes that were present in both sexes (trichoid sensillum subtypes I and II, campaniform sensillum, primary rhinarium subtypes I and II), there were four types or subtypes with size differentiation between both sexes. The trichoid sensilla subtype I of males were significantly longer than those of sexual females. The campaniform sensilla, and primary rhinaria subtypes I and II of males were significantly larger than those of sexual females. The differences for trichoid sensilla subtype I and primary rhinaria subtype I were dramatic, with 44% and 51% enlargement in males, respectively (Table 1). Thus, sensilla sizes were universally enlarged in males compared to the sexual females.

### 3.4. Secondary Rhinaria Contributed the Most to Sexual Dimorphism

We investigated the distribution and number of antennal sensilla in both sexes and found that the distribution and quantity of trichoid sensilla subtype II, campaniform sensilla, and primary rhinaria subtypes I and II were not sexually dimorphic. Four trichoid sensilla subtype II were distributed only on the tip of the PT of the FL4 (Figure 4A,B and Figure 5A). One campaniform sensillum was located on the dorsal side of the pedicel (Figure 4A,C and Figure 5B). One primary rhinarium subtype I was located on the distal end of the penultimate segment (FL3) (Figure 4A,D and Figure 5C). On the terminal segment, one primary rhinarium subtype II was positioned on the distal end of the BA adjacent to the PT (Figure 4A,D and Figure 5D).

The most widely distributed sensilla, subtype I trichoid sensilla, were sparsely scattered over the surface of each antennal segment in both sexes (Figure 4A,E). They showed sexual dimorphism in terms of quantity with males having many more trichoid sensilla subtype I than the sexual females (T = 9.473, *p* < 0.01) (Figure 5E). Moreover, the largest difference was observed for the secondary rhinaria, which were absent in sexual females. These rhinaria, which were present only on males’ antennae, were distributed on the ventral side of the FL1–3 antennal segment (Figure 4F–K). There were 59.47 ± 0.88 secondary rhinaria in total on the males’ antennae with 38.73 ± 0.65, 12.73 ± 0.45, and 8.00 ± 0.20 on FL1–3, respectively (Figure 5F). Therefore, secondary rhinaria contributed the most to antennal sexual dimorphism.

## 4. Discussion

In this study, the antennae and sensilla were evaluated in both sexes of *S. heraclei*. The flagellum length contributed to the majority of the sexual dimorphism of the antennae. Almost all sensillum types were enlarged in males. With respect to distribution and quantity, trichoid sensilla subtype II, campaniform sensilla, and primary rhinaria subtypes I and II exhibited similar profiles in the two sexes, while trichoid sensilla subtype I and secondary rhinaria showed sexual dimorphism. The antennae of males bore more trichoid sensilla subtype I than those of sexual females. Secondary rhinaria were enriched on the antennae of males, while they were not detected on the antennae of sexual females.

Aphids belong to Pterygota, whose antennae are annulated antennae which consist of three segments: a scape, a pedicel, and a flagellum with many flagellomeres [15,28]. The antennae of *S. heracei* were longer in males than in sexual females mainly due to flagellum length differentiation between the sexes. A similar result was also found in *S. avenae* [16]. In *Macrosiphoniella sanborni*, the apterous viviparous female had longer antennae than the alate [29]. Therefore, the antennal length is related to morph or sex. The scape and the pedicel of the annulated antennae have muscles to support and control the direction of the antennae. While the flagellum does not have muscle and cannot move autonomously, it might function in perception of the environment [15]. In *S. heracei*, the flagellum accounts for most of the antennal length and performs the main function of antennae. On the one hand, a longer flagellum has a larger surface to bear more sensilla; on the other hand, it has a larger search radius in environmental perception. However, the sensilla are sparse on the antennae of aphids, and the flagellum might function as radar-like feelers, and flagellum length might represent the radius of this radar. As receptors in the process of sexual information exchange, longer antennae or flagellum could perceive a greater environmental range and could help males better meet their demand of seeking and locating host plants or mates during flight.

The trichoid sensilla subtype II, campaniform sensilla, and primary rhinaria types had similar profiles in the two sexes of *S. heraclei*. Notably, all three sensillum types had a fixed number and position. Similar results were found in all the morphs of most aphid species studied, such as *Ac. pisum*, *M. persicae*, *S. avenae*, *Ma. Sanborni*, and *Megoura viciae* [16,17,18,29,30]. The conserved characters of these three sensilla indicate that they have common roles in the aphid group. There are four trichoid sensilla subtype II on the tip of FL4, the main function of which is contact chemoreception. Aphids always touch the leaves with the tip of their antennae when moving on the leaves, so trichoid sensilla subtype II are thought to participate in the selection of aphid feeding sites on host plants [31,32]. Primary rhinaria are deduced to function in host plant seeking and interspecific alarming. There is one primary rhinarium subtype I located on the distal end of FL3 and one primary rhinarium subtype II positioned on distal end of the BA adjacent to the PT of FL4. Primary rhinaria subtype I are involved in the detection of plant volatiles and primary rhinaria subtype II are involved in the detection of plant volatiles and alarm pheromones [19,33]. There is one campaniform sensillum on the dorsolateral side of the pedicel. Campaniform sensilla are mechanoreceptive sensilla that detect cuticle stresses [31]. It is speculated that this sensillum type could coordinate the movements of the scape and pedicel to ensure that antennal chemoreceptors are oriented in the correct direction to function [16,31]. The lack of a difference in these sensilla between sexes or morphs suggests they have functions related to individual survival.

Although the trichoid sensilla subtype I were distributed along almost the whole length of antennae in both sexes, they showed sexual dimorphism in quantity. The number of trichoid sensilla subtype I in males was significantly higher than that in sexual females. A similar result was found in *S. avenae* where males had more trichoid sensilla subtype I than did sexual females [16]. Trichoid sensilla subtype I are mechanoreceptors with tactile functions [31]. The antennae with more trichoid sensilla subtype I have a greater perception ability. In addition, the sensilla were universally enlarged in males of *S. heraclei*. This male bias phenomenon was also found in *M. persicae* with longer trichoid sensilla subtype I and trichoid sensilla subtype II in males [18]. Therefore, sensillum size is potentially a feature with a sex or morph bias. Larger sensilla have more perception units to sense more info chemicals, and they potentially have a higher sensitivity and perception efficiency. The quantity and size of sensilla are potentially the main factors affecting their perception sensitivity.

Secondary rhinaria were different in terms of distribution and quantity between the sexes in *S. heraclei*. The secondary rhinaria were present on males’ FL1–3 antennal segment but absent from females’ antennae. These results were consistent with those reported in *M. persicae* and *S. avenae* [16,18]. The number of secondary rhinariaon each segment of FL1–3 was 40, 14, and 9 in *S. heraclei*, 17–26, 20, and 8 in *M. persicae*, and 34, 11, and 9 in *S. avenae*, respectively. The segment nearer to the distal end of the antennae has fewer secondary rhinaria. For *Ac. pisum*, although the antennae of males bore secondary rhinaria with a distribution similar to that in *S. eracleid*, *M. persicae*, and *S. avenae*, the sexual females still had some residual secondary rhinaria only on FL1 [17]. Additionally, males of *A. glycines* have been reported to bear numerous secondary rhinaria on FL1–3. There were 23–29, 17–28, 10–14, and 4–6 secondary rhinaria on FL1–4 of the antennae in *A. gossypii*. Information on sexual females in *A. gossypii* and *A. glycines* is unavailable [19,20]. Enrichment of secondary rhinaria is so common in male aphids that it is deduced to play an important role. Secondary rhinaria bearing many pores on the surface are thought to be typical olfactory chemoreceptors [34,35]. In most aphids, the male recognizes a sexual female via sex pheromones [11]. Therefore, secondary rhinaria in males are proposed to detect sex pheromones and this assumption has been verified by single-cell recordings (SCRs) in *A. fabae*, *Ac. pisum*, *Phorodon humuli* and *Cryptomyzus galeopsidis* [33,36,37,38].

Secondary rhinaria are present in males of *S. heraclei*, *M. persicae*, and *S. avenae* and absent from the sexual females of these species. Coincidentally, the males of these species have wings while the sexual females do not. It seems likely that secondary rhinaria are related to the differences between alate and apterous morphs. Studies on the antennae of viviparous females with wing dimorphism have indicated that the secondary rhinaria are correlated with the wing [17,29,30,32]. Therefore, the difference in secondary rhinaria between males and sexual females can likely be attributed to their wing forms. However, in addition to alate males, *Ac. pisum* has another strain with apterous males whose antennae bear many more secondary rhinaria than those of sexual females [17]. Additionally, the males of *Aphis verbasci* [4], *Brachycolus cucubali* [5], and *P. brachychaeta* [39] are the apterous morph. In these species, there are still many secondary rhinaria on FL1–3 of males but none on those of sexual females. Enrichment of secondary rhinaria on the antennae of both winged and wingless males indicate that the difference in the profiles of secondary rhinaria is largely attributable to sex. Thus, future research should focus on studying secondary rhinaria with regard to the mechanisms underlying chemical communication between sexual aphids to assist in pest control.

## Figures and Tables

**Figure 1 insects-14-00468-f001:**
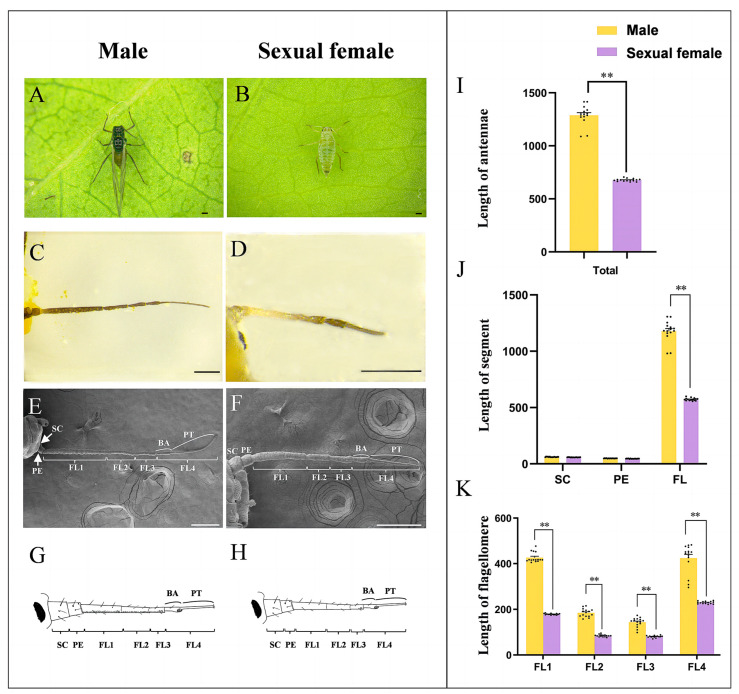
The antennae exhibit dramatic differences between sexes in *Semiaphis heraclei* (Takahashi). (**A**,**B**) Adult male (**A**) and sexual female (**B**). (**C**,**D**) Light microscope image of the antennae of male (**C**) and sexual female (**D**). (**E**,**F**) Scanning electron image of the antennae of male (**E**) and sexual female (**F**). (**G**,**H**) Diagram of the antennae of male (**G**) and sexual female (**H**). (**I**) Length of the antennae of males and sexual females. (**J**) Length of each segment of the antennae of males and sexual females. (**K**) Length of each flagellomere of the flagellum of males and sexual females. SC: scape; PE: pedicel; FL: flagellum; BA: base part; PT: processus terminalis. The bar chart shows the mean ± SEM of the data; the data of males and sexual females were analyzed by *t* test; Each dot in the bar chart represents the index value of each repeated measurement. **: *p* < 0.01. Scale bar = 200 μm.

**Figure 2 insects-14-00468-f002:**
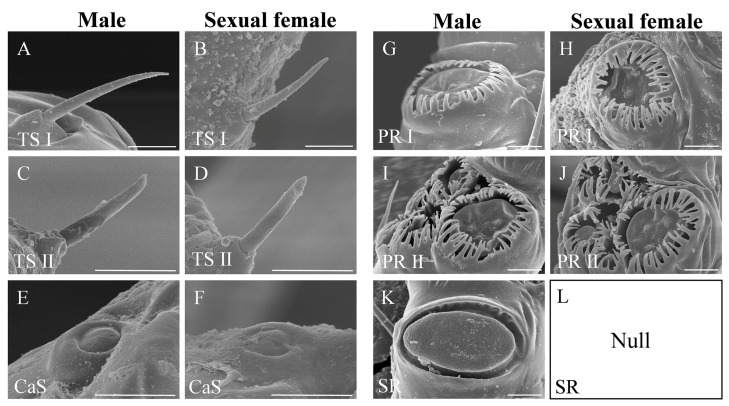
The type of antennal sensilla or rhinaria identified on the antennae of both sexes in *Semiaphis heraclei* (Takahashi). (**A**,**B**) Trichoid sensillum subtype I (TS I) of male (**A**) and sexual female (**B**). (**C**,**D**) Trichoid sensillum subtype II (TS II) of male (**C**) and sexual female (**D**). (**E**,**F**) Campaniform sensillum (CaS) of male (**E**) and sexual female (**F**). (**G**,**H**) Primary rhinarium subtype I (PR I) of male (**G**) and sexual female (**H**). (**I**,**J**) Primary rhinarium subtype II (PR II) of male (**I**) and sexual female (**J**). (**K**) Secondary rhinarium (SR) of male. (**L**) Secondary rhinarium (SR) was absent from the antennae of sexual female. Scale bar = 5 μm.

**Figure 3 insects-14-00468-f003:**
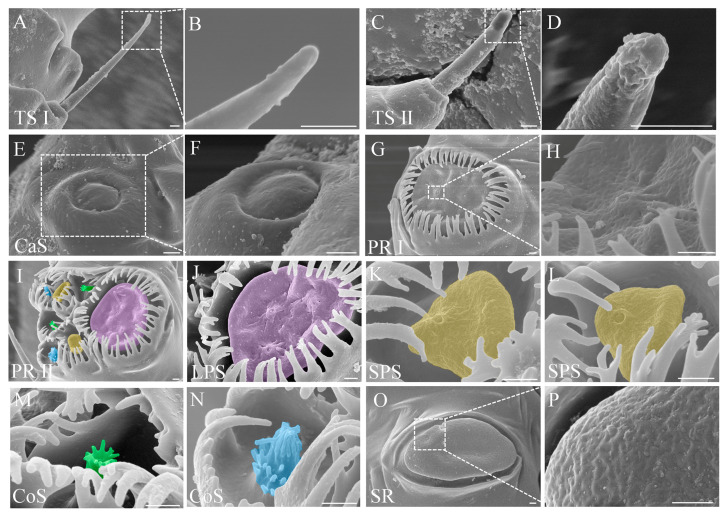
The morphology of antennal sensilla or rhinaria in male *Semiaphis heraclei* (Takahashi). (**A**,**B**) Trichoid sensillum subtype I (TS I). (**C**,**D**) Trichoid sensillum subtype II (TS II). (**E**,**F**) Campaniform sensillum (CaS). (**G**,**H**) Primary rhinarium subtype I (PR I). (**I**) Primary rhinarium subtype II (PR II): large placoid sensillum (purple), two small placoid sensilla (yellow), and four coeloconic sensilla (with two sensory cones radiating outward (green) and the other two closed (blue)). (**J**) Large placoid sensillum (LPS). (**K**,**L**) Small placoid sensilla (SPS). (**M**) Coeloconic sensillum (CoS) with sensory cones radiating outward. (**N**) Coeloconic sensillum (CoS) with closed sensory cones. (**O**,**P**) Secondary rhinarium (SR). Scale bar = 1 μm.

**Figure 4 insects-14-00468-f004:**
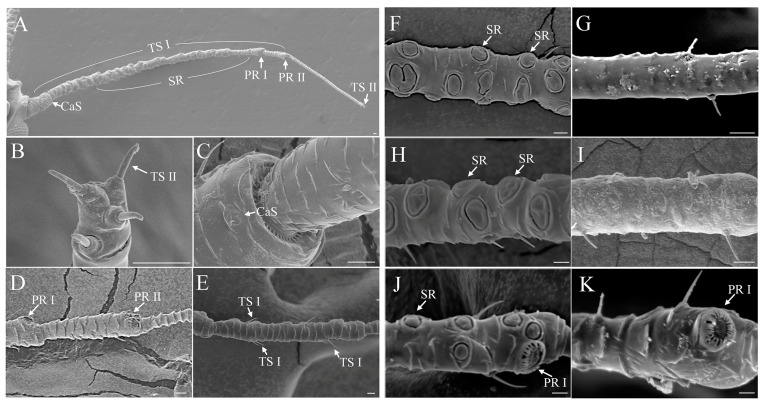
Distribution of sensilla on the antennae in *Semiaphis heraclei* (Takahashi). (**A**) Overview of the position of sensilla on the antenna of male. (**B**) Four trichoid sensilla subtype II on the tip of the antenna. (**C**) A campaniform sensillum on the distal end of the pedicel. (**D**) A primary rhinarium subtype I and a primary rhinarium subtype II on the third and fourth flagellomeres. (**E**) Many trichoid sensilla subtype I on the second flagellomere of the flagellum. (**F**,**G**) Ventral side of the first flagellomere of males (**F**) and sexual females (**G**). (**H**,**I**) Ventral side of the second flagellomere of males (**H**) and sexual females (**I**). (**J**,**K**) Ventral side of the third flagellomere of males (**J**) and sexual females (**K**). TS I: trichoid sensillum subtype I; TS II: trichoid sensillum subtype II; CaS: campaniform sensillum; PR I: primary rhinarium subtype I; PR II: primary rhinarium subtype II; SR: secondary rhinarium. Scale bar = 10 μm.

**Figure 5 insects-14-00468-f005:**
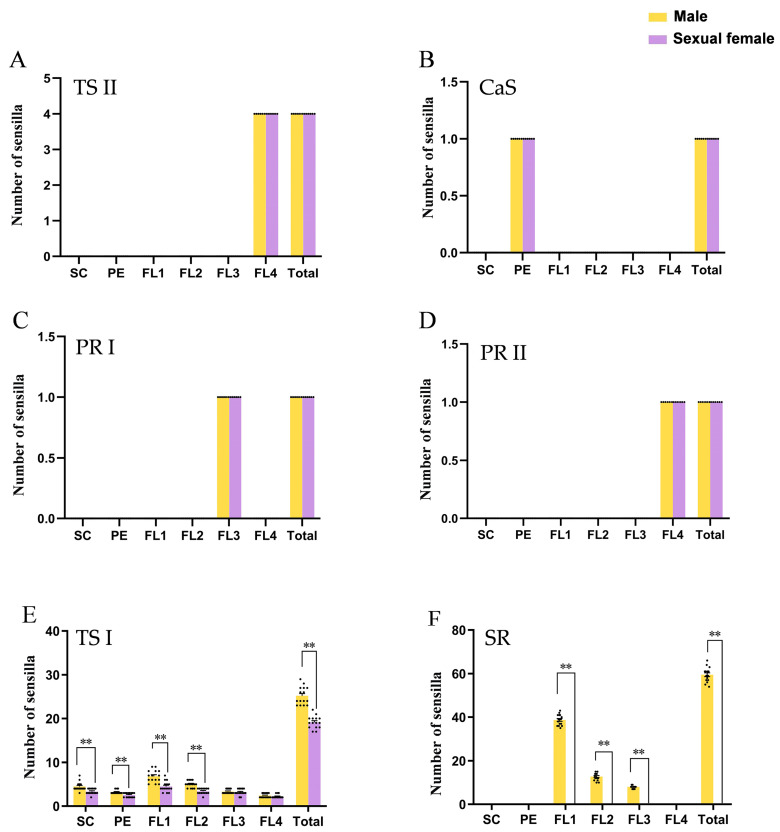
Number of sensilla on each section of antennae in *Semiaphis heraclei* (Takahashi). (**A**) Trichoid sensillum subtype II (TS II). (**B**) Campaniform sensillum (CaS). (**C**) Primary rhinarium subtype I (PR I). (**D**) Primary rhinarium subtype II (PR II). (**E**) Trichoid sensillum subtype I (TS I). (**F**) Secondary rhinarium (SR). The bar chart shows the mean ± SEM of the data; the data of males and sexual females were analyzed by *t* test; Each dot in the bar chart represents the index value of each repeated measurement. **: *p* < 0.01.

**Table 1 insects-14-00468-t001:** The shape, length, and diameter of antennal sensilla in *Semiaphis heraclei* (Takahashi).

Sensilla	Sensilla within Rhinaria	Wall	Socket	Tip	Index	Male	Sexual Female	t Value	*p* Value
TS I	/	Smooth	Raised	Slender	L BD	14.40 ± 0.49 ** 1.15 ± 0.03	10.01 ± 0.34 1.14 ± 0.02	7.39 0.18	<0.01 0.86
TS II	/	Smooth	Raised	Blunt and grooved	L BD	6.13 ± 0.14 1.15 ± 0.01 **	5.95 ± 0.06 1.12 ± 0.04	1.15 −3.13	0.26 <0.01
CaS	/	Smooth	Wide	Button-like	D	3.22 ± 0.04 **	3.02 ± 0.04	3.47	<0.01
PR I	PS	Rough	Wide	Discoid	LD SD	14.61 ± 0.32 ** 11.72 ± 0.24 **	9.70 ± 0.17 8.57 ± 0.23	13.61 9.32	<0.01 <0.01
	The whole	Rough	Wide	Discoid	LD SD	20.97 ± 0.20 ** 15.09 ± 0.24	18.55 ± 0.14 13.75 ± 0.21	9.79 4.12	<0.01 <0.01
PR II	LPS	Rough	Wide	Discoid	LD SD	9.69 ± 0.20 ** 6.66 ± 0.27 **	6.39 ± 0.08 5.70 ± 0.15	15.25 3.09	<0.01 <0.01
	SPS	Rough	Wide	Discoid	LD SD	3.22 ± 0.06 2.51 ± 0.07 **	3.14 ± 0.06 2.43 ± 0.04	0.95 −3.73	0.35 <0.01
	CoS	Smooth	Sunken	Finger-like	D	5.19 ± 0.06 **	4.66 ± 0.04	7.57	<0.01
SR	PS	Rough	Wide	Discoid	LD SD	13.71 ± 0.76 8.49 ± 0.27	/ /	20.35 36.18	<0.01 <0.01

Abbreviations TS I: trichoid sensillum subtype I; TS II: trichoid sensillum subtype II; CaS: campaniform sensillum; PR I: primary rhinarium subtype I; PR II: primary rhinarium subtype II; SR: secondary rhinarium; PS: placoid sensillum; LPS: large placoid sensillum; SPS: small placoid sensillum; CoS: coeloconic sensillum; L: length; D: diameter; BD: basal diameter; LD: long axis diameter; SD: short axis diameter. Footnotes: Mean ± SEM; the data of male and sexual female were analyzed by *t* test; **: *p* < 0.01; /: the structure was absent.

## Data Availability

The data presented in this study are available in the article.

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
