# Peer review of "Secondary Rhinaria Contribute to Major Sexual Dimorphism of Antennae in the Aphid Semiaphis heraclei (Takahashi)"

_insects, 2023, doi:10.3390/insects14050468_

Round 1

Reviewer 1 Report

Song et al. compared the antennal sensilla between male and sexual female aphid by using electron microscopy and found the secondary rhinaria only existing on the male antennae and sensillum trichoid subtype I was more abundant on the male antennae. These sex-biased antennal sensilla were referred to play important roles in the perception of sex pheromones. The work was well done and the manuscript was well written. The only weakness is only morphology of the antennal sensilla was presented in this study. It would be great if the authors could conduct some eletrophysiological and behavioral test to show the sex-biased roles of the sex-biased sensilla.

Author Response

Dear Reviewer,

Thanks for your review comments. My colleagues and I appreciate your positive and constructive comments to our manuscript. The comments and suggestions are not only great encouragement but also very valuable for us to improve the quality of our manuscript. Responses to your comments and suggestions by point-to-point are marked in red color.

Best regards,

Kun Guo

Institute of Medicinal Plant Development,

Chinese Academy of Medical Science/Peking Union Medical College,

Beijing 100193, China

[email protected]

Song et al. compared the antennal sensilla between male and sexual female aphid by using electron microscopy and found the secondary rhinaria only existing on the male antennae and sensillum trichoid subtype I was more abundant on the male antennae. These sex-biased antennal sensilla were referred to play important roles in the perception of sex pheromones. The work was well done and the manuscript was well written. The only weakness is only morphology of the antennal sensilla was presented in this study. It would be great if the authors could conduct some eletrophysiological and behavioral test to show the sex-biased roles of the sex-biased sensilla.

Response: We thank you for the good suggestion. We understand that eletrophysiological and behavioral test could reveal the role of the sex-biased sensilla. We design our project into two parts. In the present study, we mainly focused on the morphology of sensilla. The second part would be the funciton of sex-biased antennal sensilla. Actually we have started to conduct electrophysiological and behavioral test which would be a part of another story.

Reviewer 2 Report

   The reviewer has read with interest the manuscript entitled as “Secondary rhinaria contribute to major sexual dimorphism of antennae in the aphid Semiaphis heraclei (Takahashi)” submitted by Ke-Xin Song, Jiang-Yue Wang, Hai-Li Qiao, Hong-Shuang Wei, Kun Guo, and Chang-Qing Xu to a science Journal INSECTS. The authors investigated with SEM external structures of antennae and antennal sensilla of the aphid Semiaphis heraclei (Takahashi) and found the sexual dimorphism in the length of antenna, flagellum and flagellomere and the distribution and number of antennal sensilla in the aphid. Furthermore, they discussed these characteristics comparing with other aphid species. Their findings are expected to be useful in understanding of chemical communication between aphids and in pest control. However, the manuscript includes some problems as follows.

1.       Figure 1 A - D are poor photographs. Please replace more clear photographs or figures.

2.       Though the authors used the high resolution SEM, many presented SEM photographs are low magnified and poorly resolved ones. Some specimens are contaminated with some substances. The reviewer recommends the authors to replace these photographs with better ones.  

3.       The nomenclature of sensilla is not adequate. When sensilla are called in Latin style, for example, trichoid sensillum is called ‘sensillum trichodeum’ (singular) or ‘sensilla trichodea’ (plural) (lines 153-154). In English style ‘trichoid sensillum’ and ‘trichoid sensilla’ are commonly used.

4.       The singular and plural styles should be distinguished even in abbreviated words. For example, ‘sensillum trichoid (ST)’ is singular but this was treated as plural in line 167. Similar usage was found in many sentences.

5.       The rhinarium was divided to primary and secondary rhinaria in the manuscript. Do the authors know the exact meanings of these words (primary and secondary)? Please check up ‘primary and secondary’ and ‘first and second”. The word ’secondary’ means less important compared to the word ‘primary’.

6.       Several photographs are used doubly; Fig 2 A = Fig 3 A, Fig 2 E = Fig 3 E, Fig 2 C=Fig 3 C, Fig 2 K=Fig 3 O. This is not good!

7.       In Figure 5 the legends are not correspondent to the graphs.

8.       There are many careless mistakes in the manuscript. The reviewer recommends the authors to ask an English native speaker to copyedit the manuscript, after the authors’ extensive revision.

   As the reviewer is not a native English speaker, he refrains from the comments on English of the manuscript.

Author Response

Dear Reviewer,

Thanks for your review comments. My colleagues and I appreciate your positive and constructive comments to our manuscript. The comments and suggestions are not only great encouragement but also very valuable for us to improve the quality of our manuscript. Responses to your comments and suggestions by point-to-point are marked in red color.

Best regards,

Kun Guo

Institute of Medicinal Plant Development,

Chinese Academy of Medical Science/Peking Union Medical College,

Beijing 100193, China

[email protected]

The reviewer has read with interest the manuscript entitled as “Secondary rhinaria contribute to major sexual dimorphism of antennae in the aphid Semiaphis heraclei (Takahashi)” submitted by Ke-Xin Song, Jiang-Yue Wang, Hai-Li Qiao, Hong-Shuang Wei, Kun Guo, and Chang-Qing Xu to a science Journal INSECTS. The authors investigated with SEM external structures of antennae and antennal sensilla of the aphid Semiaphis heraclei (Takahashi) and found the sexual dimorphism in the length of antenna, flagellum and flagellomere and the distribution and number of antennal sensilla in the aphid. Furthermore, they discussed these characteristics comparing with other aphid species. Their findings are expected to be useful in understanding of chemical communication between aphids and in pest control. However, the manuscript includes some problems as follows..

Point 1: 1 Figure 1 A - D are poor photographs. Please replace more clear photographs or figures.

Response 1: We thank you for raising this point. Due to our negligence, the photographs of Figure 1 were compressed and their resolution had been reducted automatically when being inserted into the main text of manuscript,. We are sorry for that. We replaced these photographs with new ones of higer resolution. And the original photographs would be uploaded in a separate compressed file (*.zip) to the editor.

Point 2: Though the authors used the high resolution SEM, many presented SEM photographs are low magnified and poorly resolved ones. Some specimens are contaminated with some substances. The reviewer recommends the authors to replace these photographs with better ones.

Response 2: We thank you for raising this point. As the photographs of Figure 1 in point 1, the SEM photographs were compressed when inserted in to the manuscript. We are sorry for our carelessness. We have replaced these photographs with new ones of higer resolution.There are many wax glands secreting wax power on the body surface of this aphid (Semiaphis heraclei) especially on sexual females. The antennae of this aphid are tiny and soft. The wax power and the antennae adhere so tightly that it is hard to remove the power completely without hurt the antennae. The contamination that the reviewer mentioned might be the wax power on the antennae.

Point 3:The nomenclature of sensilla is not adequate. When sensilla are called in Latin style, for example, trichoid sensillum is called ‘sensillum trichodeum’ (singular) or ‘sensilla trichodea’ (plural) (lines 153-154). In English style ‘trichoid sensillum’ and ‘trichoid sensilla’ are commonly used.

Response 3: We thank you for raising this mistake we made.We are sorry for that. We adopt the nomenclature of sensilla in English style and revised all sensilla names in the manuscript .

Point 4:The singular and plural styles should be distinguished even in abbreviated words. For example, ‘sensillum trichoid (ST)’ is singular but this was treated as plural in line 167. Similar usage was found in many sentences.

Response 4: We are sorry for this mistake we made. In order to distinguish the singular and plural styles more clearly, we replaced the abbreviated name of all sensilla with their full names in the manuscript.
Point 5:The rhinarium was divided to primary and secondary rhinaria in the manuscript. Do the authors know the exact meanings of these words (primary and secondary)? Please check up ‘primary and secondary’ and ‘first and second”. The word ’secondary’ means less important compared to the word ‘primary’.

Response 5: The primary and secondary rhinaria are morphological terms in the aphid study. They were commonly used in morphological studies such as Wu et al, (2022); Kanturski et al, (2020), Zhang and Zhang, (2000); and Du et al, (1995). Both the primary and secondary rhinaria of aphid are important olfactory organs and play different function in environment perception.

Point 6: Several photographs are used doubly; Fig 2 A = Fig 3 A, Fig 2 E = Fig 3 E, Fig 2 C=Fig 3 C, Fig 2 K=Fig 3 O. This is not good!

Response 6: We thank you for your careful review. We replaced the doubly used photographs of Figrue 2 with new ones in Line 157.

Point 7: In Figure 5 the legends are not correspondent to the graphs.

Response 7: We thank you for your careful review. We revised the figure legends of Figure 5 to“Number of sensilla on each section of antennae in Semiaphis heraclei (Takahashi). (A) Trichoid sensillum subtype II (TS II). (B) Campaniform sensillum (CaS). (C) Primary rhinarium subtype I (PR I). (D) Primary rhinarium subtype II (PR II). (E) Trichoid sensillum subtype I (TS I). (F) Secondary rhinarium (SR). The bar chart shows the mean ± SEM of the data; the data of males and sexual females were analyzed by t test; **:P < 0.01.” in Line 249-253

Point 8: There are many careless mistakes in the manuscript. The reviewer recommends the authors to ask an English native speaker to copyedit the manuscript, after the authors’ extensive revision.

Response 8: Thank you for your careful review. We are very sorry for the mistakes in this manuscript and inconvenice they caused in your reading. After extensive revision, this manuscript has been copyedited by an English native speaker via lauguage editing service of MDPI (English editing ID: English-65852).

Reviewer 3 Report

This is an excellent paper on the ultrastructure of male and female antennae of the celery aphid.  Line 35 requires some structure change, the easiest is "Aphids are" rather than aphid is. Line 322 sentence also needs some modification "males of them have wing", the easiest change is to add an s to wing. The paper makes a good effort to tie the flagellum morphology of the male antennae to female reception. However, in the end, this remains only an educated guess. With this new knowledge how would you suggest it be used in the pest control of the insect? 

Author Response

Dear Reviewer,

Thanks for your review comments. My colleagues and I appreciate your positive and constructive comments to our manuscript. The comments and suggestions are not only great encouragement but also very valuable for us to improve the quality of our manuscript. Responses to your comments and suggestions by point-to-point are marked in red color.

Best regards,

Kun Guo

Institute of Medicinal Plant Development,

Chinese Academy of Medical Science/Peking Union Medical College,

Beijing 100193, China

[email protected]

This is an excellent paper on the ultrastructure of male and female antennae of the celery aphid.

Point 1: Line 35 requires some structure change, the easiest is "Aphids are" rather than aphid is.

Response 1: Thank you for your suggestion, we revised this sentence to “Aphids are devastating pests of agricultural crops. ”in Line 40

Point 1: Line 322 sentence also needs some modification "males of them have wing", the easiest change is to add an s to wing.

Response 2: Thank you for your suggestion, we revised this sentence to “Coincidentally, the males of these species have wings while the sexual females do not.” in Line 342-343.

Point 3: The paper makes a good effort to tie the flagellum morphology of the male antennae to female reception. However, in the end, this remains only an educated guess. With this new knowledge how would you suggest it be used in the pest control of the insect?

Response 3: We thank you for rasing this point. Understanding the mechanism underlying chemical communication between sexual aphids could be helpful for exploring olfactory stimulus-based male trapping techniques. Usually, aphids use antennal sensilla to percept sex pheromone. Secondary rhinaria of both winged and wingless males is largely attributable to sex and they are potentially the main sensilla to percept sex pheromones. Secondary rhinaria are suggested to be focused in the research on mechanism underlying chemical communication between sexual aphids to assist in pest control. In the end of MS, we add “Thus, secondary rhinaria are suggested to be focused on in future research on the mechanisms underlying chemical communication between sexual aphids to assist in pest control.” In Line 354-356.

Round 2

Reviewer 1 Report

I have no additional comments. Seems the authors prefer to overlook the weakness of the paper.